# CAR T-Cells in Multiple Myeloma Are Ready for Prime Time

**DOI:** 10.3390/jcm9113577

**Published:** 2020-11-06

**Authors:** Paula Rodríguez-Otero, Felipe Prósper, Ana Alfonso, Bruno Paiva, Jesús F. San Miguel

**Affiliations:** Clínica Universidad de Navarra, Centro de investigación médica aplicada (Cima), CIBERONC, IDISNA, 31008 Pamplona, Spain; fprosper@unav.es (F.P.); aalfonso@unav.es (A.A.); bpaiva@unav.es (B.P.); sanmiguel@unav.es (J.F.S.M.)

**Keywords:** relapse and refractory multiple myeloma, B-cell maturation antigen, chimeric antigen receptor T cell (CAR T-cell) therapy, immunotherapy

## Abstract

The survival of patients with multiple myeloma (MM) has been dramatically improved in the last decade thanks to the incorporation of second-generation proteasome inhibitors (PI), immunomodulatory drugs (IMID), and, more recently, anti-CD38 monoclonal antibodies (MoAb). Nevertheless, still, a major proportion of MM patients will relapse, underscoring the need for new therapies in this disease. Moreover, survival in patients failing the current standard of care regimens (including PI, IMIDs, and anti-CD38 MoAb), which is now defined as triple-class refractory, remains dismal, and new drugs with different mechanism of action are needed. B-cell maturation antigen (BCMA)-targeted therapies and in particular chimeric antigen receptor T cell (CAR T-cell) treatment have emerged as promising platforms to overcome refractoriness to conventional drugs. In this manuscript, we review the current available data regarding CAR T-cell therapy for MM, with a special focus on target selection, clinical results, limitations, and future strategies.

## 1. Introduction

The treatment landscape of multiple myeloma (MM) patients has substantially changed in the last decade thanks to the incorporation of several new drugs and combinations, including second-generation proteasome inhibitors (PI) (carfilzomib, ixazomib), second-generation immunomodulatory drugs (IMIDs) (pomalidomide), and anti-CD38 monoclonal antibodies (MoAb) (daratumumab). This has led to a significant improvement in survival times both in the frontline and relapse setting [1]. Nevertheless, still, a major proportion of MM patients will relapse, underscoring the need for new therapies in this disease. Moreover, the current standard of care in MM treatment includes continuous treatment as a major driver for improved survival. This continuous therapy approach from the first line to end-stage disease determines the development of resistant clones in the setting of continuous treatment, leading to drug refractoriness and decreased survival upon relapse.

Anti-CD38 monoclonal antibodies, and in particular daratumumab, have been one of the major drivers of this increase in MM survival and are now incorporated in all the treatment phases in combination with other backbones. A recent retrospective study focused on the outcome after anti-CD38 MoAb failure has clearly showed that median overall survival (OS) from refractoriness to CD38-MoAb is significantly shortened (8.6 months) and particularly for those patients that are refractory to five different drugs (penta-refractory patients), with a median OS of only 5.6 months [2]. Therefore, there is a clear need for new therapies, with different mechanism of action, for the management of patients with relapse and refractory MM (RRMM).

Immunotherapy-based approaches, and in particular T-cell based therapies, have emerged in the last years as a promising platform to overcome refractoriness to conventional drugs. In line with this, clinical experience using chimeric antigen receptor T-cells (CAR T-cells) or T-cell engager antibodies is rapidly increasing with promising results already reported in end-stage MM.

In this article, we will review the available data regarding CAR T-cell therapy for MM, with a special focus on target selection, clinical results, limitations, and future strategies.

## 2. Engineered T-Cells in MM: CAR T-Cells

Until recently, adoptive cell therapy in myeloma was restricted to bone marrow transplant or the anecdotal use of tumor-infiltrating lymphocytes (TILs). Clinical experience with TILs in MM is scanty and is mainly restricted to the work of Borrello et al. using marrow-infiltrating lymphocytes (MILs) [3]. Nonetheless, progress in gene engineering technologies has simplified the generation of specific antitumor T-cells, overcoming many of the practical barriers that have limited the wide dissemination of adoptive cell therapy using TIL cells. Genetically redirecting a T-cell’s specificity toward a patient’s cancer cell can be accomplished in two ways: (1) a cloned T-cell receptor (TCR) conferring tumor recognition is inserted into circulating lymphocytes, and (2) T-cells are transduced with a chimeric antigen receptor (CAR).

CARs are engineered fusion proteins that contain (1) an extracellular antigen-binding domain composed of a single-chain variable fragment (scFv) derived from an Ab that confers recognition to a tumor-associated antigen; and (2) an extracellular hinge/spacer and a transmembrane domain that is linked in tandem to (3) intracellular signaling motifs capable of T cell activation. First-generation CARs only contained the CD3 portion of the TCR but had suboptimal clinical activity with lower persistence. The second and third-generation CARs significantly improved the activity through the incorporation of costimulatory molecule domains such as CD28, 4-1BB, or OX40. Fourth-generation CARs express additional molecules to enhance CAR-T cell efficacy, such as inducible interleukin. So far, the strongest clinical data derive from second-generation CAR T-cells based on one single costimulatory molecule (either CD28 or 4-1BB) (Figure 1).

## 3. Target Selection

Selecting the appropriate antigen is a critical step in CAR design both to optimize efficacy whilst preventing on target-off tumor toxicity. The ideal target for CAR development is one whose expression is restricted to the tumor cell; however, most-tumor associated target antigens are also expressed, albeit at low-to-intermediate levels, on normal tissues [4]. Different antigens are expressed in the surface of plasma cells (Figure 2) and have been used for CAR-T design (Table 1); however, B-cell maturation antigen (BCMA) has been the focus of attention in the past recent years.

B-cell maturation antigen (BCMA), also named TNFRSF17 or CD269, is a membrane bound of the tumor necrosis factor receptor (TNFR) superfamily [5]. Ligands for BCMA include B-cell activating factor (BAFF) and a proliferation-inducing ligand (APRIL), of which APRIL has a higher affinity for BCMA [6,7]. The expression of BCMA is restricted to the B-cell compartment, being expressed in clonal and polyclonal plasma cells, as well as in a small subset of normal memory B cells. BCMA is overexpressed in MM preclinical models and patients and has been reported to play a role in myeloma cell growth, chemoresistance, and microenvironment immunosuppression [8,9].

BCMA expression in myeloma patients is almost universal [7], but the density of antigen expression in the cell is highly variable and significantly lower as compared to other surface antigens such as CD38. Moreover, the membrane expression of BCMA may vary over time due to the shedding of soluble BCMA (sBCMA) after cleavage by a γ-secretase enzyme [10]. sBCMA could be followed in the serum of MM patients, and it is still unclear whether it could impair the efficacy of BCMA-targeting drugs.

Other targets that are being exploited for the development of CAR T-cells are as follows:(1)CD38 is expressed with high intensity in clonal plasma cells and has been shown to be a promising target for the treatment of MM. However, CD38 is also expressed, although with less intensity, on normal hematopoietic cells, such as red blood cells, natural killer cells, and other tissues, increasing the risk of on-target off-tumor toxicity [11]. There is already preclinical evidence showing an activity of CD38-targeting CAR T-cells, and several clinical trials are ongoing.(2)CD138 (Syndecan 1) is expressed in normal and clonal plasma cells and also on normal tissues such as epithelial cells potentially inducing “on target–off tumor” toxicity. However, in a preclinical study using CD138-directed CAR T-cells, no epithelial toxicity was observed [12]. The same experience was noted in a clinical report with five patients treated with CD138 directed CAR T-cell in China, and a phase 1 trial is ongoing [11].(3)The orphan G protein–coupled receptor, class C group 5 member D (GPRC5D), is expressed ubiquitously in malignant bone marrow plasma cells, hair follicles, and variably in the lung tissue [13]. Interestingly, expression in MM cells is 500 to 1000 times that found on normal cells [11]. CAR T-cells targeting GPRC5D have demonstrated promising preclinical activity [13], and clinical strategies targeting GPRC5D, particularly using bispecific antibodies, are under evaluation.(4)Signaling lymphocyte activation molecule F7 (SLAMF7 or CS1) is widely expressed on plasma cells, as well as subsets of normal B and T-cells, natural killer (NK) cells, monocytes, and dendritic cells, and it is already a target used in MM therapy with the antibody elotuzumab [11,14,15]. CAR T-cells targeting SLAMF7 have shown encouraging preclinical activity; however, SLAMF7 expression in lymphocyte subsets raised the problem of the specific fratricide of SLAMF7^+/high^ target cells by SLAMF7-CAR T-cells, although SLAMF7^−/low^ was preserved and able to remain viable [16]. Clinical trials evaluating SLAMF7-targeted CAR T-cells are ongoing.(5)CD19-directed CAR T-cell therapy is approved for the treatment of B acute lymphoblastic leukemia and diffuse large B cell lymphoma. CD19 is typically absent on the dominant multiple myeloma cell population, but it may be present on a minor subset with unique myeloma-propagating properties [17,18]. CD19-directed CAR T-cell therapy has been tested in MM patients after autologous stem cells therapy with interesting results [18] and also in combination with BCMA-directed CAR T-cells in small series of patients with promising data [19].(6)The activating receptor NKG2D (natural-killer group 2, member D) and its ligands play an important role in the NK, γδ+, and CD8+ T-cell-mediated immune response to tumors. Ligands for NKG2D are rarely detectable on the surface of healthy cells and tissues, but they are frequently expressed by tumor cell lines and in tumor tissues, which makes them attractive targets for CAR development [20]. NKG2D ligand-directed CAR T-cells have been evaluated both in preclinical and clinical settings (albeit a small number of patients) with promising results [21]. NKG2D ligand-directed CAR NK cells are also under development [22].

## 4. Clinical Data Using CAR T-Cells in MM

Clinical data using CAR T-cells in myeloma is still scanty. No CAR T-cell product has been yet approved or is available for the treatment of MM patients outside the context of clinical trials. In the recent years, the number of clinical trials evaluating CAR T-cell therapy in multiple myeloma is continuously increasing. Although different constructs and targets are being evaluated, still, BCMA-directed CAR T-cells (Table 2) accumulate the most robust evidence with one product (idecabtagene vicleucel, bb2121, ide-cel) likely to be approved by the U.S. Food and Drug Administration.

The first clinical trial reporting efficacy using a BCMA-targeting CAR T was presented in 2015 and conducted at the National Cancer Institute [28,29]. The BCMA CAR T construct used in this study contained a murine anti-BCMA single-chain variable fragment, hinge, and transmembrane regions from human CD8α, the CD28 costimulatory molecule, and the CD3z T-cell activation domain [28]. A total of 27 patients were included and 24 infused, with 16 patients at the higher dose level of 9 × 10^6^ CAR T-cells/kg. The median number of prior treatments in the highest dose level was 9.5. The overall response rate (ORR) among these 16 patients was 81% with a median event-free survival of 31 weeks, including six patients with ongoing response at the moment of the publication. Peak CAR^+^ cell levels occurred between 7 and 14 days after CAR–BCMA T-cell infusion for all patients and correlated with disease response. CAR–BCMA T-cell toxicity was mild at lower doses but substantial at the highest dose level. The cytokine release syndrome (CRS) of any grade was present in 94% of the patients (15/16) with six out of 16 patients developing grade 3-4 CRS and 31% of patients requiring treatment with tocilizumab.

Another BCMA-directed CAR T construct has been codeveloped by the University of Pennsylvania and Novartis and tested in a phase I trial [30]. The structure contained a fully human scFV and a 4-1BB costimulatory domain, which were transferred using a lentiviral vector-based technology. Twenty-nine patients were included in three sequential cohorts: cohort 1, 1 × 10^8^ to 5 × 10^8^ CART–BCMA cells alone; cohort 2, cyclophosphamide (Cy) 1.5 g/m^2^ plus 1 × 10^7^ to 5 × 10^7^ CART–BCMA cells; cohort 3, Cy 1.5 g/m^2^ plus 1 × 10^8^ to 5 × 10^8^ CART–BCMA cells. Twenty-five patients were infused (four patients were never infused due to rapid MM progression). The median number of prior lines was 7, with 44% of patients being penta-refractory. Interestingly, BCMA–CAR T-cells were administered in an outpatient research unit over 3 days as split-dose intravenous infusions. Twenty-one out of 25 patients received the full CAR T dose, whilst four patients received 40% of the dose due to early CRS onset. The ORR for the three cohorts was 48% and was higher in the third cohort (ORR 64%). The median progression-free survival (PFS) was 65, 57, and 125 days for each cohort, respectively. Cytokine release syndrome was seen in 88% of the infused patients and was grade 3 or higher in eight (32%) subjects, all of whom were treated at the 1 × 10^8^ to 5 × 10^8^ dose. Median time to CRS onset was 4 days, with a median duration of 6 days. Neurotoxicity was seen in eight (32%) patients and was mild in five subjects, with three patients presenting with grade 3–4 encephalopathy. Responses were significantly associated with peak expansion as well as with persistence over the first 28 days. No association was found between expansion or response and age, years from diagnosis, number of prior lines, TP53 mutation status, or penta-refractory disease. Even more, pre-apheresis treatment, baseline serum BCMA concentration, MM cell BCMA intensity, or bone marrow plasma cell infiltration were neither associated with expansion or response. Contrariwise, a higher number of CD27^+^CD45RO^−^CD8^+^ T-cells within the leukapheresis product correlated with a most robust in vivo expansion and better clinical response.

A third BCMA–CAR construct initially named LCAR-B38M (China) subsequently JNJ-68284528, and currently, Ciltacabtagene Autoleucel (cilta-cel) has been evaluated in three different early phase trials (Legend-2, CARTITUDE-1, and CARTIFAN-1). Cilta-cel is a structurally differentiated CAR T-cell therapy with a 4-1BB costimulatory domain and two B-cell maturation antigen (BCMA)-targeting domains. Cilta-cel has been given breakthrough therapy designation by the U.S. Food and Drug Administration and PRIME (PRIority MEdicines) status by the European Medicines Agency.

The first trial reported using this CAR construct was the Legend-2 trial, a phase I, first-in-human (FIH) study conducted at multiple centers in China, each with their own lymphodepletion protocol and timing for CAR T-cell administration. Data from 57 patients treated at a single institution in China (The Second Affiliated Hospital of Xi’an Jiaotong University) were first published in 2018 [31] and updated recently with a median follow-up of 25 months [27]. Lymphodepleting therapy was cyclophosphamide alone (300 mg/m^2^ days −5, −4 and −3), and CAR T-cells were administered in three split doses, 20% of the total dose on day 1, 30% on day 3, and the remaining 50% on day 7. The median dose infused was 0.5 × 10^6^ cells/kg (range 0.07–2.1 × 10^6^). The median number of prior lines was 3, with very few patients with prior exposure to the second generation of novel agents, or anti-CD38 monoclonal antibodies, reflecting a less heavily pretreated population. The overall response rate (ORR) was 88%, with a significant proportion of patients achieving complete response (CR) (74%) and minimal residual disease (MRD) negativity (68%) using eight-color flow with a sensitivity of 10^−6^. Responses were rapid (median time to first response was 1.1 month) and deepened over time. The median duration of response (DOR) was 27.0 months (95% Confidence Interval (CI) 14.3–NE), and the median progression-free survival (PFS) and overall survival (OS) for all treated patients was 19.9 months (95% CI, 9.6–31.0) and 36.1 months (95% CI 26.4—not estimated (NE)), respectively. Both PFS and OS were longer for patients achieving CR, 28.2 months (95% CI, 19.9–NE), and not reached (NR) (95% CI, 35.0–NE), respectively. Safety was manageable with 90% of patients presenting CRS, which was grade 3-4 in only 4 patients (7%). Median time to onset of CRS was 9 days, with a median duration of 9 days [32]. Tocilizumab was used in 45% of the subjects. Neurotoxicity was infrequent (2%). Clinical responses did not correlate with BCMA expression, and in most patients, (71%) LCAR-B38M CAR T-cells were not detectable in peripheral blood at 4 months.

The same dual-epitope binding BCMA–CAR construct has been evaluated in the phase 1b/2 CARTITUDE-1 study, which was conducted in the U.S. Eligible patients had to have received at least three prior lines of therapy with exposure to prior proteasome inhibitors, immunomodulatory agents, and antiCD38 monoclonal antibodies. A total of 35 patients were enrolled and 29 were infused with a median age of 60 years. The median number of prior lines were 5, ranging from 3 to 18, with 97% of patient refractory to the last line of therapy and 86% triple-class refractory. Fludarabine (30 mg/m^2^, 3 days) and cyclophosphamide (300 mg/m^2^, 3 days) were given as conditioning treatment and CAR T-cells were infused at day 1 as a single infusion. The median administered dose of cilta-cel was 0.73 × 10^6^ (0.52–0.89 × 10^6^) CAR^+^ viable T-cells/kg. ORR at the last data cut-off, with a median FUP of 11.5 months, was 100% with a significant proportion of subjects achieving stringent complete remission (86%). Responses were durable and deepened over time with a median time to first response and CR of 1 and 3 months, respectively. Median PFS has not been reached yet, with a 9-month PFS of 86% (95% CI, 67–95). The overall safety profile was manageable with CRS and cytopenias being the most frequent adverse events. CRS was presented in 93% of the patients with grade ≥3 in only two patients. Median time to CRS onset was 7 days with a median duration of 4 days. Neurotoxicity was infrequent (10%) and mostly low grade. Only one patient developed grade ≥3 immune effector cell-associated neurotoxicity syndrome (ICANs). Cytopenias were common and generally grade 3 or higher. Grade ≥ 3 neutropenia and thrombocytopenia were presented in 100% and 69% of the patients, respectively. Prolonged cytopenias, lasting beyond 60 days, were uncommon, and median time to grade 3–4 neutropenia and thrombocytopenia recovery was 1.6 weeks (95% CI, 1.3–1.9) and 5.3 weeks (95% CI, 2.4–8.1), respectively [26].

Interestingly, and in contrast with prior experience with other BCMA CAR-T constructs, both peak cilta-cel expansion and CAR+ T-cell persistence did not correlate with clinical response in this trial. In 18 out of 28 patients, the number of CAR+ T-cells were below the limit of quantitation at 3 months of follow-up, with several patients losing the CAR T-cells in periphery as early as by day 28. Nonetheless, despite early CAR T-cell loss in several patients, responses were durable and deepen over time, suggesting that the loss of peripheral persistence may not be associated with relapse for all CAR T-cell therapies.

Idecabtagene vicleucel (ide-cel; bb2121) is the most advanced BCMA-directed CAR T-cell therapy investigated in relapsed and refractory multiple myeloma patients. Ide-cel is comprised of a murine extracellular single-chain variable fragment (same as in the National Cancer Institute study) attached to a human CD8 ∝ hinge and transmembrane domain fused to the T-cell cytoplasmic signaling domains of the 4-1BB and CD3-ζ chain, in tandem. Ide-cel has been granted Breakthrough Therapy designation (BTD) by the U.S. Food and Drug Administration, and PRIority MEdicines (PRIME) designation and validation of its Marketing Authorization Application (MAA) by the European Medicines Agency for relapsed and refractory multiple myeloma, and it is likely to be the first CAR T-cell therapy approved for the treatment of RRMM patients. The phase 1 study included 33 patients with RRMM who have received at least three prior lines of treatment, with prior exposure to proteasome inhibitors and IMIDs or double-refractory disease. For the dose–escalation portion of the study, BCMA expression on 50% or more of marrow plasma cells on immunohistochemical assay was required, although this criterion was eliminated in the expansion phase. In addition, prior daratumumab exposure and refractoriness to the last line of treatment were required during the expansion phase. Ide-cel was administered as a single infusion at doses of 50 × 10^6^, 150 × 10^6^, 450 × 10^6^, or 800 × 10^6^ CAR+ T-cells in the dose–escalation portion of the study and 150 × 10^6^ and 450 × 10^6^ CAR+ T-cells in the expansion phase. Results for the first 33 consecutive patients infused were published in 2019 [33]. Hematologic adverse events were the most common events with grade 3 or higher neutropenia in 85% and thrombocytopenia in 45%. CRS was present in 25 patients (76%), which was grade 1 or 2 in 23 patients (70%) and grade 3 in two patients (6%). Neurologic toxicity occurred in 14 patients (42%) and were of grade 1 or 2 in 13 patients (39%). The ORR was 85%, including 15 patients (45%) with complete response. The median PFS was 11.8 months (95% CI, 6.2–17.8). The peak CAR-T cell expansion was associated with clinical response, and CAR T-cells persisted up to 1 year after the infusion.

The pivotal phase 2 KarMMa study of ide-cel recruited 158 patients, of whom 140 underwent leukapheresis and 128 were infused with three different doses of ide-cel: 150 × 10^6^ (*n* = 4), 300 × 10^6^ (*n* = 70), and 450 × 10^6^ (*n* = 54). Eligible patients had three or more prior lines of treatment, prior exposure to proteasome inhibitors, IMIDs, and anti-CD38 monoclonal antibodies and were refractory to the last line of treatment. The baseline characteristics of the patients reflected a very advanced stage of a heavily pretreated myeloma population with a median number of six prior lines and 84% of triple-class refractory subjects. Moreover, 51% of the patients presented high tumor burden defined as bone marrow infiltration ≥50% and 39% had extramedullary disease. Most patients (88%) received bridging therapy during CAR T-cell manufacturing, and only 4% responded. Both primary and secondary endpoints of the trial were met. The ORR across all infused patients was 73% with a CR rate of 33%. There was a clear dose–response relation with higher ORR (82%) and CR (39%) among patients treated at the higher dose level (450 × 10^6^). The median time to first response was 1.0 months, and the median time to CR was 2.8 months. The median PFS in all treated patients was 8.8 months (95% CI, 5.6–11.6), and it was 12.0 months for patients treated with the higher dose of 450 × 10^6^ CAR+ T-cells. The median PFS was superior in patients achieving CR with a median PFS of 20.2 months (95% CI, 12.3–NE) as compared to 5.4 m (95% CI, 3.8–8.2) for those patients in partial response. As seen in other CAR-T trials, both cytopenias and CRS were the most common adverse events. CRS was reported in 84% of all treated patients (96% at the 450 × 10^6^ CAR+ T cell dose), with grade 1–2 CRS in 78% and grade ≥3 in <5% of patients. The median time to CRS onset was 1 day with a median duration of 5 days. Tocilizumab and steroids were used in 67% and 19% of the patients, respectively. Neurotoxicity was uncommon (18% of the patients treated) and mostly grade 1–2. On the opposite, cytopenias were common and were not related to the cells’ dose. The incidence of grade ≥ 3 neutropenia and thrombocytopenia was 89% and 52%, respectively. The median time to recovery from grade ≥ 3 neutropenia or thrombocytopenia was 2 months (95% CI, 1.9–2.1) and 3 months (95% CI, 2.1–5.5), respectively. Delayed recovery (>1 month) of grade ≥3 neutropenia and thrombocytopenia were seen in 41% and 48% of the patients, respectively. Overall survival is still immature, with 66% of patients censored overall. The median OS was 19.4 months (95% CI, 18.2–NE). In this study, peak CAR T-cell expansion did correlate with clinical outcome, and cell expansion correlated with longer PFS in the higher dose levels [23].

In an attempt to increase the duration of remission, several strategies are being evaluated to increase the proportion of T-cells with a central memory phenotype based on experiences showing better clinical outcomes in these patients [34] and prolonged CAR-T cell persistence [11]. So far, two different alternatives are being studied in phase 1 clinical trials: one incorporating a PI3K inhibitor during T cell culture (bb21217) and a second one using a 1:1 CD4:CD8 ratio in the culture (Orvacabtagene-autoleucel, orva-cel, JCARH125; EVOLVE study).

The phase 1 trial with bb21217 included 36 patients in the last updated report with ORR across three different dose levels (150, 300, and 450 × 10^6^) of 60% and a manageable safety profile [24]. Longer follow-up is needed to assess if treatment with bb21217 translates into sustained CAR T-cell persistence and durable clinical responses.

More mature results have been recently presented with orvacabtagene autoleucel (orva-cel) in the phase 1/2 EVOLVE study [25]. Eligible patients had three or more prior lines of treatment, including proteasome inhibitors, IMIDs, and anti-CD38 MoAb and were refractory to the last line of treatment. Sixty-two patients were included and treated at the higher dose levels of 300 × 10^6^, 450 × 10^6^, and 600 × 10^6^ CAR+ T-cells. The median age was 61 years. The median number of prior lines was six with 94% of triple-class refractory patients. Extramedullary disease was identified in 23% of the patients using baseline PET-CT scan. The ORR across all three doses was 92% with 36% of CR. Although follow-up is still short at the higher-dose levels (3.9 and 2.3 months for 450 and 600 × 10^6^ CAR+ T-cells), the responses seem durable. The median PFS was 9.3 months for the 300 × 10^6^ CAR+ T-cells cohort and not yet reached for the 450 and 600 × 10^6^ CAR+ T-cell cohorts. The safety profile was manageable and overall comparable to other trials. CRS was present in 89% of the patients with only two patients experiencing grade ≥3 CRS. The median time to CRS onset was 2 days, and the median duration was 4 days. Neurotoxicity was present in only 13% of the subjects and mostly grade 1–2. Cytopenias were common. Neutropenia grade ≥3 was seen in 90% of the patients and thrombocytopenia grade ≥3 was seen in 47%. Prolonged cytopenias were also common. Sixty-seven percent of the patients had grade ≥3 cytopenia at day 29 and 35% had grade ≥3 cytopenia at month 2. The 600 × 10^6^ CAR+ T-cells dose has been selected as the recommended phase 2 dosing, and the expansion phase of this study is now enrolling.

## 5. Limitations of Current Approaches and Potential Avenues

### 5.1. Peak Expansion and Response

Despite significant efficacy with high ORR and CR rates in the different BCMA-directed CAR-T trials, with survival significantly superior to that reported with real-world therapies [35], patients continue to relapse, and no plateau is seen in the survival curves. Mechanisms of resistance and relapse following CAR T-cell therapy in MM are poorly understood. CAR-T cell expansion and peak CAR-T cell levels do correlate with response across different trials [28,30,33]. In the KarMMa [23] study, peak ide-cel vector copies were significantly higher in responders as compared to non-responders, and a similar experience was published with the UPenn BCMA CAR T-cell [30]. However, recent data using cilta-cel in the phase 1b/2 CARTITUDE-1 study fail to show this correlation. In this study, ORR was 100%, and responses were independent of peak CAR-T expansion [36]. It is still unclear whether this is related to the small number of patients treated or to the differences in CAR-T design with a dual-binding domain and potential higher affinity, and more follow-up is needed.

### 5.2. Antigen Escape

Antigen escape or downmodulation has also been described as a potential mechanism of progression after CAR T-cell treatment. This has been demonstrated after CD 19-directed CAR T-cell therapy in acute lymphoblastic leukemia and has also been observed in some cases in myeloma [37]. Indeed, decreased levels of BCMA expression as well as BCMA downmodulation has been described in patients after BCMA CAR T therapy [30], and some strategies are ongoing evaluating combination treatment with γ-secretase inhibitors (GSI) to increase surface BCMA levels and decrease soluble BCMA. Preliminary data from one phase I study evaluating the combination of BCMA CAR-T and JSMD194 (a GSI) have been recently presented. Treatment with GSI efficiently increased the level of BCMA expression on the plasma cell with promising efficacy (ORR of 100% among 10 patients treated). However, the incidence of neurological toxicity was significant (70%) [38]. It is important to note that the results so far reported suggest no correlation between BCMA expression levels and response, although in the ide-cel KarMMa study, there was a trend showing higher BCMA receptor density for patients with a very good partial response (VGPR) or CR/stringent CR.

On the other hand, BCMA loss is infrequent after BCMA CAR T-cell therapy. Indeed, the majority of the relapsing patients in the ide-cel study (15 out of 16), the largest dataset so far, showed BCMA-expressing CD138+ cells at the moment of relapse; indeed, BCMA antigen loss was found in only 4% (3 out of 71) of patients in progression with one subject with confirmed genomic loss due to biallelic deletion of the TNFRSF17 gene [39]. Even so, strategies to mitigate antigen escape are needed. This may include new target selection, dual targeting, or bispecific CAR T-cells. A bispecific BCMA-CD19 CAR T-cell has been evaluated in a first-in-human study in China. Five patients were infused without any severe adverse events, and only three out of five patients experienced grade 1 CRS. All patients responded with one patient achieving an stringent CR, 3 achieving VGPR, and 1 achieving partial response. A phase 1 trial is ongoing to further evaluate this strategy [40]. Another approach using separate infusions of CD19- and BCMA-targeted CAR T-cells has been evaluated in RRMM patients in a phase 1 trial in China. A total of 28 patients were treated with an ORR of 92.6% and a CR rate of 40.7%. CRS was reported in all patients with grade 3–4 in 32.1% of the subjects [19]. Another approach is the use of bispecific CAR T-cells. BM38 CAR-T incorporates the anti-CD38 and anti-BCMA single-chain variable fragment in tandem plus 4-1BB signaling and CD3 zeta domains. In the first-in-human study conducted in China, 16 patients were treated with an ORR of 87.5%, CR rate of 50%, 9-month PFS of 75%, and a manageable safety profile. Although these results are promising, a longer follow-up is needed to determine if dual targeting is able to improve on the results of current BCMA CAR-T products.

### 5.3. CAR T-Cell Persistence and Duration of Response

A failure in the long-term functional persistence of CAR T-cells has been postulated as a mechanism of disease progression. Loss of CAR T-cell peripheral persistence may precede relapse in some patients, and this has been seen with myeloma and other malignancies. In the KarMMa study, durable ide-cel persistence (≥6 months) was seen in 29 patients out of 49 evaluable (59%), but only four patients had CAR-T detectable by month 12 [23]. On the other hand, in the CARTITUDE-1 study, despite a quite early loss of cilta-cel in the peripheral blood in several patients, responses were maintained and even deepen over time, reflecting the variability between studies and the different behavior seen with the different CAR T constructs [36].

As discussed above, several strategies [24,25] are being developed to increase the proportion of long-lived T-cells with a memory phenotype in the infused product, since this has been associated with improved outcomes [34] and longer CAR T-cell persistence [11]. In addition to those already discuss in this article, another interesting approach is the use of novel transduction methods, such as the nonviral PiggyBac transposon-based DNA-delivery system [41]. This method has a shorter manufacturing time, lower cost, and results in a higher number of memory stem cells. It also has a very large cargo capacity that can potentially accommodate more CARs into a CAR T-cell product in the future [42]. One such CAR product being studied in MM is the P-BCMA-101 [43]. This CAR-T is under evaluation in a phase 2 study [44]. Results from the phase 1 dose escalation reported an ORR of 63% among 19 evaluable patients at five different dose levels (from 0.75 × 10^6^ up to 15 × 10^6^ cells/kg) with a median PFS of 9.5 months (NCT03288493) [45].

### 5.4. Combination Therapies

Finally, another strategy to improve the duration of response and PFS is to combine CAR T-cell with other therapies that may enhance their activity. Combinations with immunomodulatory drugs, such as lenalidomide or daratumumab or even with checkpoint inhibitors have been postulated, and some studies will start soon evaluating some of these strategies in the clinic.

## 6. Practical Considerations

When thinking about CAR T-cell treatment, there are some aspects that need to be taken into account. The first aspect is manufacturing time. The time lag between apheresis and infusion of autologous CAR T-cell remains a challenge, especially for those patients with aggressive or rapidly progressive disease. In our personal experience, the drop-out rate was 20% mainly due to complications associated with disease progression, underlying the importance of adequate patient selection. Industry and academic centers have the challenge to deliver the CAR T-cells in the shortest time possible. This problem can be solved using allogeneic CAR T-cells. These products have the advantage of being “off the shelf” and derived from healthy unrelated donors, without the need of patient apheresis and avoiding potential manufacturing failures. Indeed, dose can be titrated or repeated if needed. Moreover, a high number of CAR T-cells can be produced and cryopreserved from a single donor, reducing the cost and allowing for the potential standardization of T-cell characteristics. BCMA-directed allogeneic CAR T-cells are under evaluation in several clinical trials. To reduce the potential for graft-versus-host disease, these CAR T-cells have been genetically edited to limit T cell receptor (TCR)-mediated immune responses. This can be accomplished either using transcription activator-like effector nucleases (TALEN) [46] or CRISPR/Cas 9 technology. For example, in one allogeneic BCMA-directed CAR T-cell (CTX120), CRISPR/Cas 9 is used to insert the CAR construct precisely into the TCR alpha constant (TRAC) locus, and to eliminate the TCR. Moreover, the class I major histocompatibility complex (MHC I) is also knock out using CRSPR/Cas 9 [47]. A phase I trial using CTX120 in RRMM is ongoing (NCT04244656).

During the manufacturing time (4–7 weeks), patients can receive “bridging therapy” to slow disease progression and maintain each patient’s clinical condition. Bridging therapy is typically individualized to the patient and adapted to previous treatments, time frame until infusion, disease’s characteristics, and pre-existing toxicities. In the KarMMa study, 88% of the subjects received bridging therapy, and only 4% of the patients responded. This underscores the importance of patient selection to reduce the risk of patients not infused due to rapid clinical deterioration, death, or severe organ damage [48].

Finally, safety is another important point to take into consideration. Thus far, the clinical experience suggests that toxicity is generally manageable. CRS is present in 80–90% of patients across all trials but is generally grade 1 or 2 with less than 10% of patients developing grade ≥ 3 CRS. Neurotoxicty is unfrequent and generally mild (grade 1 or 2). On the other hand, cytopenias are very common and can be persistent with some products. Mechanisms underlying the development of cytopenias are poorly understood and include the toxicity of prior regimens, lymphodepleting chemotherapy, and inflammatory response after CAR T-cell therapy. Supportive care and prophylactic measures are needed in some patients, and prolonged cytopenias may indeed limit subsequent therapy at the moment of disease progression.

## 7. Conclusions

BCMA CAR T-cell therapy is showing impressive results in an end-stage myeloma population, but relapses still occur. Multiple questions are still unanswered, and several strategies are under investigation to improve current results. Adequate patient selection and earlier use in the course of the disease may surely impact the long-term outcome of CAR T-cell therapy; however, issues such as cost, competition with other immunotherapies in particular bispecific antibodies, and BCMA antibody–drug conjugates (such as belantamab mafodotin already FDA and EMA-approved), and the limitation of administration requiring specialized centers may limit the widespread use of this therapy today. Nevertheless, the positive initial results in patients that would not have other treatment options, and the rapidly evolving field, convert CAR-T cell therapy in one of the most promising therapeutic tools in the MM armamentarium.

## Figures and Tables

**Figure 1 jcm-09-03577-f001:**
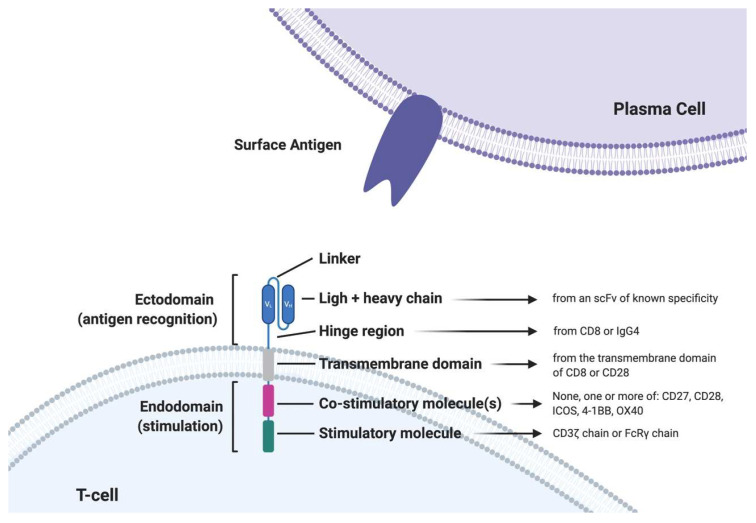
Structure of a second generation chimeric antigen receptor T cell (CAR T-cell). scFV: single chain variable fragment; FcRγ: Fc gamma receptor.4-1BB: also called CD137 or TNFRS9, activation-induced costimulatory molecule; iCOS: inducible T-cell costimulator.

**Figure 2 jcm-09-03577-f002:**
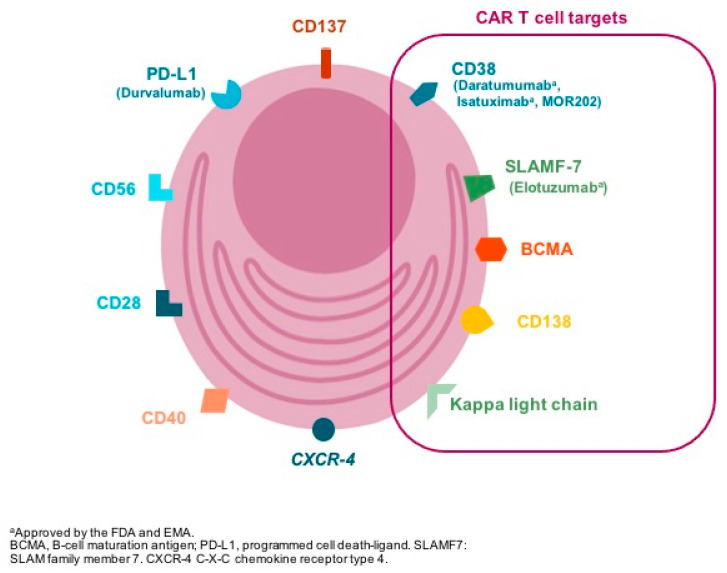
The figure represents some of the membrane targets expressed in the clonal plasma cell surfable that are suitable for drug development. In brackets, antibodies targeting specific antigens are included (i.e., daratumumab: anti-CD38 monoclonal antibody). CAR T cell: chimeric antigen receptor T cell; FDA: U.S. Food and Drug Administration; EMA: European Medical Agency.

**Table 1 jcm-09-03577-t001:** Summary of potential targets for CAR T-cell development including patterns of expression and reference to ongoing/planned clinical trials.

Antigen	Expression in Plasma Cells	Function	Off-Target Expression	Ongoing Trials
BCMA	Universal in plasma cells	Membrane bound of the TNFR superfamily	Restricted to B-cell compartment	NCT04309981 NCT03430011 NCT03288493 NCT04181827 NCT04133636 NCT04093596 NCT04196491 NCT03601078 NCT03651128 NCT04244656 NCT04394650
CD38 (Syndecan 1)	Overexpressed in multiple myeloma cells	As a receptor, CD38 can bind CD31 in T cells, activating them. As an enzyme, it catalyzes the synthesis and hydrolysis of cyclic ADP-ribose.	Normal hematopoietic cells: red blood cells, NK cells	NCT03464916 NCT03473496 NCT03767751
GPRC5D	Universal in plasma cells	Not yet been determined	Hair follicle and lung tissue	NCT04555551
SLAMF7	Overexpressed in multiple myeloma cells	Mediates activating or inhibitory effects in NK cells	Normal B and T-cells, NK-cells, monocytes, and dendritic cells	NCT04499339
CD19	Rarely detected in plasma cells	Involved in B-cell maturation	All B-lineage cells	NCT04194931 NCT04182581 NCT03767725 NCT04236011 NCT03455972 NCT04162353 *
NKG2D	Not expressed in plasma cells	Important role in the NK, γδ+, and CD8+ T-cell-mediated immune response to tumors	Rarely detectable on healthy cells and tissues	Under development

* All clinical trials with CD19 CAR-T are either dual CD19/BCMA CART or trials combining both CAR-T (CD19 y BCMA). CAR T cell: chimeric antigen receptor T cell; NKG2D: natural-killer group 2, member D; BCMA: B-cell maturation antigen; GPRC5D: orphan G protein–coupled receptor, class C group 5 member D; SLAMF7: Signaling lymphocyte activation molecule F7; ADP: adenosin diphosphate; NK: Natural-killer; NCT: National Clinical trial; γδ+: Gamma-delta positive T-cell.

**Table 2 jcm-09-03577-t002:** Summary of efficacy and safety of most relevant B-cell maturation antigen (BCMA)-directed CAR T-cell trials in relapse and refractory multiple myeloma (RRMM).

	Idecabtagene Vicleucel (Ide-Cel) KARMMA Study [23]	bb21217 [24]	Orvacabtagene-Autoleucel (Orva-Cel) EVOLVE Ph 1/2 Trial [25]	Ciltacabtagene Autoleucel (Cilta-Cel) CARTITUDE 1 [26]	LCAR-B38M LEGEND-2 [27]
CAR Design	Autologous, lentiviral vector 4-1BB	Ide-cel cultured with PI3Ki, to enrich memory-like T cells	Fully human (CD28/41BB). 1:1 CD4:CD8 ratio	2 BCMA-targeting single chain antibody	2 BCMA-targeting single chain antibody (same as JNJ 4528)
Population	128 (Ph 2)	38 (Ph 1)	62 (Ph 1/2)	29 (Ph 1b/2)	57 (Ph 2)
Number of Prior lines	6	6	6	5	2
CAR T-cell Dose	150–450 × 10^6^ CAR T-cell	150–450 × 10^6^ CAR T-cell	300–600 × 10^6^ CAR T-cell	0.73 × 10^6^ CAR T-cells/kg	0.5 × 10^6^ CAR T + Cells/kg
Refractory to CD38 MoAb	94%	76%	NA	93%	1 patients
Triple-class Refractory	84%	63%	94%	86%	--
Extramedullar disease	39%	NA	23%	10%	NA
ORR (CR)	82%* (39%)	NA (33%)	92% ^#^. (29%)	100% (86%)	88% (74%)
DOR/PFS/ OS months	11.3 */12.1 */19.4	11.1 ^&^/NA/NA	NA/NA/ NA	NA/86%@9m/NA	19.9m In CR: mPFS 28.2 m
CRS (G 3/4)	96% * (6%)	66% (6%)	88% ^#^ (4%)	93% (7%)	90% (7%)
Neurotox(G3)	20% * (6%)	24% (8%)	13% ^#^ (0%)	10% (3%)	2%
Reference	(1) Munshi NC, et al. Initial KarMMa results. J Clin Oncol. 2020;38 (suppl; abstr 8503).	(2) Berdeja JG, et al. Blood (Internet). 2019 Nov 13;134 (Supplement_1):927.	(3) Mailankody S, et al. J Clin Oncol. 2020;30 (suppl; abstr 8504).	(4) Usmani SZ, et al. EHA Library. 2020. p. EP926.	(5) Chen L, et al. Blood (Internet) 2019 Nov 13;134 (Supplement_1):1858.

* Responses at 450 × 10^6^ CAR T dose levels. ^#^ Responses at 600 × 10^6^ CAR T dose level. ^&^ Responses at 150 × 10^6^ CAR T dose level. ORR: overall response rate. CR: complete response. DOR: duration of response. PFS: progression-free survival. OS: overall survival. CRS: cytokine release syndrome. G 3/4: Grade 3 or 4. Neurotox: neurotoxicity. G3: Grade 3.

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
