# Peer review of "CAR T-Cells in Multiple Myeloma Are Ready for Prime Time"

_jcm, 2020, doi:10.3390/jcm9113577_

Round 1
Reviewer 1 Report
This is a well written and comprehensive manuscript focusing on main available clinical evidences about CAR-T therapy in multiple myeloma.
Besides reviewing main clinical trials, it focuses on the main challenges and open issues related to CAR-T therapy in MM.
Minor suggestions:
- It would be useful for the reader to see a Table summarizing potential targets for CAR-T developement mentioned in paragraph 3, their patterns of expression and references to ongoing/planned clinical trials
- In paragraph 3, line 86: Syndecan 1 is CD138 and not CD38. Also CD138 (syndecan 1) has been considered as a potential target for CAR-T development in MM. Please add a comment in the paragraph.
- Lines 333-334: Zhang et al presented a bispecific tandem CD19-BCMA CAR-T (Blood 2019) and a trial is ongoing. Please include also this evidence in the paragraph.
Author Response
We thank the reviewer for all the comments and suggest. Please find below a point-by-point response the reviewer 1 (responses are in italics):
Comment 1. It would be useful for the reader to see a Table summarizing potential targets for CAR-T developement mentioned in paragraph 3, their patterns of expression and references to ongoing/planned clinical trials
We thank the reviewer for the suggestion and have added the table (Table 1) to the manuscript.
Comment 2. In paragraph 3, line 86: Syndecan 1 is CD138 and not CD38. Also CD138 (syndecan 1) has been considered as a potential target for CAR-T development in MM. Please add a comment in the paragraph.
We thank the reviewer and have amended the mistake. Information about CD138 has also been included in the manuscript.
Comment 3. Lines 333-334: Zhang et al presented a bispecific tandem CD19-BCMA CAR-T (Blood 2019) and a trial is ongoing. Please include also this evidence in the paragraph.
We thank the reviewer for the comment. This evidence has been incorporated in the manuscript.
Reviewer 2 Report
Rodriguez-Otero et al provide a comprehensive review of current literature and data of CAR T-cell therapy in myeloma. The article is well-organized and summarizes the important trials that have been recently reported and are ongoing. I only have minor suggestions to consider as follows:
-The authors describe the CAR T-cell construct but a visual graphic may be useful for the readers to conceptualize the CAR and costimulatory domains
-Line 127: Comment is made regarding anticipated approval of ide-cel later this year. I would perhaps remove the timeline and instead just state that FDA approval is expected
-Would recommend that "CAR T-cell therapy" is written the same way throughout the manuscript. Some sections write it as "CAR-T cell" or "CAR T therapy" etc.
-The title of the paper suggests that real world application of CAR T-cell therapy will be discussed. The authors comment on this in their practical considerations and conclusions. I would add some more details about the competitive landscape including specific bispecifics antibodies and also mention belantamab mafodotin since this is a current FDA approved BCMA directed therapy. I would also comment on how the authors might think CAR T-cell therapy may fit into the current therapeutic landscape and comment on other non-BCMA agents currently used in this space.
Author Response
We thank reviewer 2 for the comments and suggestions. Please find below the point-by-point response to the reviewer's comments in italics.
Comment 1. The authors describe the CAR T-cell construct but a visual graphic may be useful for the readers to conceptualize the CAR and costimulatory domains
We thank the reviewer for the suggestion and a figure (Figure 2) has been included in the manuscript.
Comment 2. Line 127: Comment is made regarding anticipated approval of ide-cel later this year. I would perhaps remove the timeline and instead just state that FDA approval is expected
We thank the reviewer for the comment and have removed the timeline reference as suggested.
Comment 3.Would recommend that "CAR T-cell therapy" is written the same way throughout the manuscript. Some sections write it as "CAR-T cell" or "CAR T therapy" etc.
We have revised the manuscript and we have homogenized the way we write CAR T-cell therapy following the reviewer’s instructions.
Comment 4. The title of the paper suggests that real world application of CAR T-cell therapy will be discussed. The authors comment on this in their practical considerations and conclusions. I would add some more details about the competitive landscape including specific bispecifics antibodies and also mention belantamab mafodotin since this is a current FDA approved BCMA directed therapy. I would also comment on how the authors might think CAR T-cell therapy may fit into the current therapeutic landscape and comment on other non-BCMA agents currently used in this space.
We thank the reviewer for this comment and we have included reference to Belantamab Mafodotin. Since this is a review focused on CAR T-cell therapy we feel it is better to leave the conclusions open without incorporating new drugs given the wide landscape of new therapies that may be incorporated in the treatment of triple-class refractory myeloma.